# RoboSVG: A Unified Framework for Interactive SVG Generation with Multi-modal Guidance

## Abstract

Scalable Vector Graphics (SVGs) are fundamental to digital design and robot control, encoding not only visual structure but also motion paths in interactive drawings. In this work, we introduce RoboSVG, a unified multimodal framework for generating interactive SVGs guided by textual, visual, and numerical signals. Given an input query, the RoboSVG model first produces multimodal guidance, then synthesizes candidate SVGs through dedicated generation modules, and finally refines them under numerical guidance to yield high-quality outputs. To support this framework, we construct RoboDraw, a large-scale dataset of one million examples, each pairing an SVG generation condition (e.g., text, image, and partial SVG) with its corresponding ground-truth SVG code. RoboDraw dataset enables systematic study of four tasks, including basic generation (Text-to-SVG, Image-to-SVG) and interactive generation (PartialSVG-to-SVG, PartialImage-to-SVG). Extensive experiments demonstrate that RoboSVG achieves superior query compliance and visual fidelity across tasks, establishing a new state of the art in versatile SVG generation. The dataset and source code of this project will be publicly available soon.

## 1 Introduction

Scalable Vector Graphics (SVGs) are the cornerstone of modern digital content creation (Samir, 2025; Eckert, 2021) and robot path control (Khafagy et al., 2025; Lu et al., 2024). Unlike raster images, SVGs are resolution-independent, compact, and human-readable, encoding graphics as structured sequences of geometric and semantic primitives. These properties make them well-suited for precise rendering, responsive design, and programmatic manipulation. Importantly, SVG paths can also be interpreted as motion trajectories, which connects SVG generation to robot drawing scenarios. Basic SVG generation corresponds to a robot creating a complete drawing directly from input text or an image. Interactive SVG generation corresponds to a robot refining or extending an existing drawing, i.e., performing interactive drawing. Despite progress in multimodal large language models (MLLMs) (Liu et al., 2023; Wang et al., 2024) and image generation (Esser et al., 2024; Batifol et al., 2025), SVG generation remains underexplored and presents a uniquely challenging task. Unlike raster image synthesis, SVG generation requires not only semantic alignment but also structural validity and visual coherence. Models must map inputs (including text, images, or partial SVGs) into sequences of graphical primitives while preserving geometric consistency. These challenges become even more demanding in interactive settings, where a model must reason about plausible continuations given incomplete information.

A key bottleneck is the absence of a large-scale, high-quality dataset that supports both basic and interactive SVG generation. To address this gap, we construct RoboDraw, a dataset of one million SVG samples tailored for robot drawing scenarios, based on MMSVG-2M (Yang et al., 2025b) and SVGX (Xing et al., 2025b) datasets. Each sample consists of a complete SVG, one partial SVG, and a textual description. RoboDraw offers two major advantages: **(i)** High semantic consistency between text descriptions and SVG content, ensuring reliable supervision; **(ii)** Simplified SVG structure with clean, consistent data, where commands are restricted, viewboxes standardized, and numerical values normalized. These properties make RoboDraw a high-quality resource that directly contributes to robust model performance. We propose four SVG generation tasks based

Figure 1: The four tasks introduced in our RoboDraw dataset, including basic generation (i.e., Text-to-SVG and Image-to-SVG) and interactive generation (i.e., PartialSVG-to-SVG and PartialImage-to-SVG).

on the RoboDraw dataset. As illustrated in Figure 1, basic generation encompasses Text-to-SVG and Image-to-SVG, which are cross-modal generation tasks that require the model to produce an SVG from either a textual description or an image. In contrast, interactive generation consists of PartialSVG-to-SVG and PartialImage-to-SVG, which closely resemble interactive drawing scenarios, where the goal is to complete a full SVG from a partial vector or a partial image.

Building on the RoboDraw dataset, we propose the RoboSVG model, a unified multimodal framework for generating versatile SVGs. RoboSVG uses multimodal guidance, which plays three critical roles: **(i)** The guidance generator extracts complementary perspectives of the input query across modalities (text, vision, and partial SVG), enabling a richer understanding of user intent. **(ii)** The guidance tools embed broad prior knowledge from powerful pretrained models. For instance, a text-to-image model can capture visual knowledge of most real-world objects for further processing. **(iii)** Numerical guidance provides both training-time optimization signals and inference-time selection criteria, ensuring that the final output SVG is structurally valid, semantically faithful, and visually coherent. Extensive experiments demonstrate that RoboSVG achieves state-of-the-art performance across both basic and interactive tasks. We benchmark against strong MLLMs (e.g., Qwen2.5-VL-72B (Bai et al., 2025), GPT-4o (Hurst et al., 2024)) as well as task-specific baselines (Xing et al., 2025b; Yang et al., 2025b; Rodriguez et al., 2025), showing consistent improvements in semantic alignment and visual quality.

In summary, we make the following contributions:

- We introduce interactive SVG generation tasks, where the goal is to generate a complete SVG from partial input (either partial SVG or partial image). To enable this, we curate the RoboDraw dataset, a large-scale dataset of one million samples. Each sample consists of a complete SVG, its corresponding partial SVGs, and paired natural language descriptions. RoboDraw defines four tasks, including basic generation (Text-to-SVG and Image-to-SVG) and interactive generation (PartialSVG-to-SVG and PartialImage-to-SVG).
- We propose RoboSVG, a unified model for generating versatile SVGs according to input queries and multimodal guidance. RoboSVG comprises two primary components: the guidance generator and the SVG generator. The guidance generator produces visual, textual, and numerical guidance, while the SVG generator synthesizes SVGs conditioned on both the input query and the multimodal guidance.
- Extensive experiments show that RoboSVG achieves state-of-the-art results on both basic and interactive SVG generation tasks. For a thorough evaluation, we also benchmark strong existing models (e.g., Qwen2.5-VL-72B, GPT-4o) on RoboDraw and SVGenius benchmarks, providing the first comprehensive baseline comparison for this problem.

## 2 RELATED WORK

**Robot Drawing.** Robot drawing (Huang et al., 2019; Frans et al., 2022; Schaldenbrand et al., 2024) encompasses systems in which a robot produces imagery (strokes, sketches, or refinements) in response to user input or interaction. E-David (Deussen, 2010) demonstrates the generation and selection of brush-stroke parameters based on input images, as well as the adjustment of these parameters via visual feedback. Multi-Pen Robust Robotic 3D Drawing (Liu et al., 2020) maps 2D

strokes onto 3D surfaces, using closed-loop planning to ensure fidelity over complex geometry. RoboCoDraw (Wang et al., 2020) stylizes a portrait input (via GANs), converts it to a sketch, and executes drawing paths optimized for robot motion. SketcherX (Song et al., 2024) pushes this further by combining vectorization techniques with interactive feedback to produce stylized portrait drawings with vector-like strokes. However, much of the previous work focuses on raster/image targets (Huang et al., 2019), brush or stroke parameter control (Schaldenbrand et al., 2023), or mapping entire sketches onto physical strokes (Chen et al., 2025a); relatively little addresses the generation of structured vector graphics (e.g., full SVG code).

**SVG Generation Models.** Early work on SVG generation explored optimization-based methods (Li et al., 2020; Jain et al., 2023; Ma et al., 2022; Xing et al., 2024), which utilize differentiable rasterization to refine paths iteratively; however, these methods are computationally expensive and often yield overly complex outputs. Hierarchical models, such as DeepSVG (Carlier et al., 2020), SVGFormer (Cao et al., 2023), and SVGBuilder (Chen & Pan, 2025), improve structural coherence by separating global layout from local details; however, they remain limited to simple icons with shallow semantics. More recently, LLM-based approaches have redefined SVG generation as code synthesis, with models such as OmniSVG (Yang et al., 2025b), LLM4SVG (Xing et al., 2025b), StarVector (Rodriguez et al., 2025), and Reason-SVG (Xing et al., 2025a). These LLM-based approaches leverage vision-language pretraining, semantic tokenization, or hybrid refinement to enhance fidelity and generalization. Building on LLM-based approaches, RoboSVG advances the field by coupling multimodal guidance with dedicated SVG generation modules.

**Multi-modal Guidance.** The frameworks can produce rich guidance to assist a variety of downstream tasks by leveraging multi-modal generators. For example, ICE (Yu et al., 2025) uses captions generated from images to enhance zero-shot generalization in classification. MCoCa (Zhao et al., 2025) integrates multi-modal contrastive features to improve cross-modal alignment. IDEA (Ye et al., 2025) incorporates image descriptions via captioning models as auxiliary inputs to CLIP-adapter frameworks. Altogether (Xu et al., 2024) focuses on re-aligning alt-text and applying editing to improve caption quality and consistency. Commonly employed tool models include text-to-image models like FLUX.1 (Batifol et al., 2025), image captioning models like Qwen-2.5-VL (Bai et al., 2025), and image editing models like Gemini-Nano-Banana (Google AI Developers, 2025). In our work, we use all of these to generate multi-modal guidance, which in turn aids in improving the performance of SVG generation.

## 3 METHODOLOGY

In this section, we present the definition of the proposed SVG generation tasks and the design of the RoboSVG model. We first describe the construction of the RoboDraw dataset and the data processing pipeline for different SVG generation tasks. Then we introduce the two key components of RoboSVG (guidance generator, SVG generator), followed by the training and inference processes.

### 3.1 TASK DEFINITION

**RoboDraw Dataset.** The RoboDraw dataset is curated to support both basic and interactive SVG generation. Each sample consists of three core components $(S_c, S_p, T_c)$, where $S_c$ is the complete SVG, $S_p$ is a partial SVG composed of several paths of $S_c$, and $T_c$ is a textual description of $S_c$. The complete SVG $S_c$ can be rendered into a complete image $I_c$, and the partial SVG $S_p$ can be rendered into a partial image $I_p$. Our data is primarily sourced from MMSVG-2M (Yang et al., 2025b) and SVGX (Xing et al., 2025b). Based on these open-source datasets, we curate a high-quality SVG dataset containing one million samples. All SVGs are normalized to a $200 \times 200$ viewbox, with drawing commands restricted to $\{M, L, C, Q, A, Z\}$. To ensure quality, we only retain samples where the CLIPScore between $T_c$ and $I_c$ exceeds 20. Moreover, numerical values in $S_c$ are standardized to integers where possible, with absolute coordinates for point positions. For evaluation, we randomly select 500 samples as the test set, while the remaining data is used for training. This means that RoboDraw consists of one million examples in the training split and 500 examples in the test split.

**SVG Generation Tasks.** To systematically study SVG generation, we formulate two task categories: basic generation and interactive generation, as shown in Figure 1. In basic generation, the

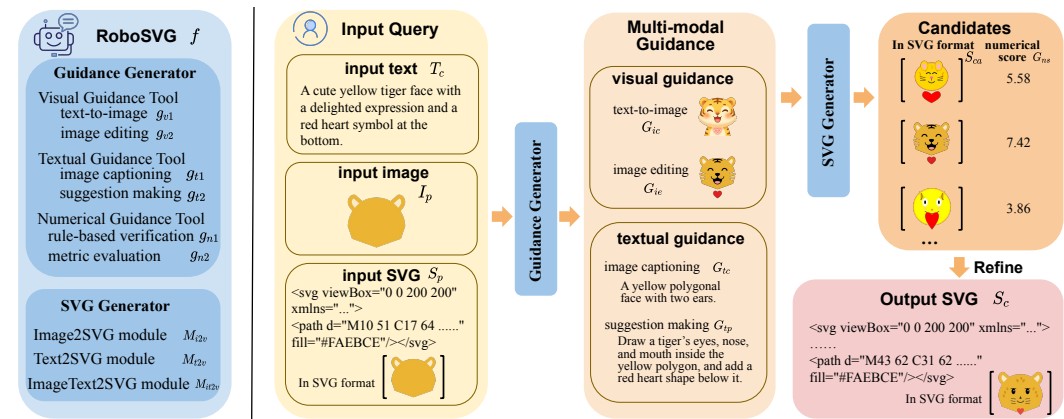

Figure 2: The architecture of the RoboSVG model. The left part illustrates the architecture of RoboSVG, consisting of the guidance generator and the SVG generator. The right part shows the workflow: the guidance generator transforms the input query into multimodal guidance, the SVG generator produces candidate SVGs, and the final output SVG is selected from these candidates.

input query typically consists of either a complete image $I_c$ or a full language description $T_c$. In interactive generation, the input query contains partial information, such as a partial SVG $S_p$ or partial image $I_p$, accompanied by a textual description $T_c$. The objective of SVG generation is thus to design a framework $f$ that maps the input query to a complete SVG $S_c$, expressed as

$$S_c = f(I_c, T_c, I_p, S_p),\tag{1}$$

where one or more of $I_c, T_c, I_p, S_p$ may be empty depending on the task. Specifically, the input queries for the four tasks are defined as follows: Text-to-SVG uses full language description as $Q_{\text{T2S}} = \{T_c\}$, Image-to-SVG uses complete image as $Q_{\text{I2S}} = \{I_c\}$, PartialSVG-to-SVG uses full language description and partial SVG as $Q_{\text{PS2S}} = \{T_c, S_p\}$, while PartialImage-to-SVG uses full language description and partial image as $Q_{\text{PI2S}} = \{T_c, I_p\}$.

## 3.2 ROBOSVG MODEL

As shown in Figure 2, the RoboSVG model consists of two primary components: a guidance generator and an SVG generator. The guidance generator produces multi-modal guidance, while the SVG generator synthesizes SVGs conditioned on both the input query and the generated guidance. We further design training and inference processes that effectively leverage multi-modal guidance to enhance the SVG generation ability across different tasks. The prompts used in the guidance generator and SVG generator are listed in Appendix A.

### 3.2.1 GUIDANCE GENERATOR

The guidance generator extracts salient information from the input query and enhances it through information augmentation and cross-modal transformation, providing the SVG generator with information-rich inputs. It integrates a suite of cross-modal processing tools, which can be grouped into three categories: visual guidance tools, textual guidance tools, and numerical guidance tools.

Visual guidance tools transform the input query into visual guidance. For input text $T_c$, text-to-image models $g_{v1}$ (e.g., FLUX.1 (Batifol et al., 2025), Stable Diffusion 3 (Esser et al., 2024)) are employed to produce complete image guidance $G_{ic}$. For input images $I_p$ (or $I_c$), image editing models $g_{v2}$ (e.g., Qwen-Image-Editing (Wu et al., 2025), Gemini-Nano-Banana (Google AI Developers, 2025)) are used to modify the image into a form that better aligns with the input text $T_c$, yielding edited image guidance $G_{ie}$. When the input contains a partial SVG $S_p$, rendering tools $g_{v3}$ (e.g., CairoSVG (Kozea, 2018)) convert it into a rasterized partial image guidance $G_{ip}$. Thus, the transformation process can be expressed as

$$G_{ic} = g_{v1}(T_c), \quad G_{ie} = g_{v2}(I_p, T_c), \quad G_{ip} = g_{v3}(S_p).\tag{2}$$

In practice, visual guidance $G_{ic}$, $G_{ie}$, and $G_{ip}$ can be treated interchangeably with $I_c$ or $I_p$, since they all lie in the visual modality.

Textual guidance tools convert the input query into textual guidance. For cases with image inputs, image captioning models $g_{t1}$ are employed to generate a complete description $G_{tc}$. For interactive drawing tasks, textual suggestions $G_{tp}$ can be produced by suggestion-making models $g_{t2}$ to guide SVG generation. The textual guidance tools are typically implemented using MLLMs (e.g., LLaVA (Liu et al., 2023), Qwen-VL (Wang et al., 2024; Bai et al., 2025)). Thus,

$$G_{tc} = g_{t1}(I_c), \quad G_{tp} = g_{t2}(T_c, I_p), \tag{3}$$

where the complete description $G_{tc}$ and the textual suggestion $G_{tp}$ serve as complementary representations of the input query intent.

Numerical guidance tools evaluate the structural validity and semantic relevance of generated SVG candidates in a quantitative manner. Rule-based verification $g_{n1}$ checks whether the SVG code follows specifications and can be successfully rendered into an image. Metric-based evaluation $g_{n2}$ measures semantic alignment between SVG candidates $S_{ca}$ and the input query. For example, CLIPScore (Radford et al., 2021) assesses the similarity between a rendered SVG image and the input text $T_c$, while SSIM (Wang et al., 2004) evaluates the similarity between a rendered SVG image and the input image $I_c$. The use of numerical guidance tools can be formulated as

$$G_{nb} = g_{n1}(S_{ca}), \quad G_{ns} = g_{n2}(T_c, I_c, S_{ca}), \tag{4}$$

where $S_{ca}$ denotes the SVG candidates, $G_{nb} \in \{0, 1\}$ indicates whether the SVG code conforms to specifications, and $G_{ns}$ is a numerical score. During training, $G_{ns}$ serves as the reward for reinforcement learning (Guo et al., 2025); while during inference, it functions as a quality metric for ranking SVG candidates.

### 3.2.2 SVG Generator

The SVG generator produces complete SVGs conditioned on the input query and multimodal guidance. Depending on the input modalities, it can be categorized into three modules, i.e., Image2SVG $V_{i2s}$, Text2SVG $V_{t2s}$, and ImageText2SVG $V_{it2s}$. To analyze the impact of input modalities on SVG generation quality, we adopt a consistent backbone architecture (i.e., Qwen-2.5-VL (Bai et al., 2025)) as the base module and train it with different modality configurations, resulting in modules with varying characteristics. When addressing SVG generation tasks, we aim to fully leverage both the input query and the multimodal guidance. To this end, we reorganize and recombine the available data according to the task requirements.

**Training Process.** In RoboSVG, the guidance generator can directly leverage existing tools, whereas all modules of the SVG generator require training. To enable fair comparisons across tasks, we adopt the same multimodal large language model (MLLM) backbone to learn different functionalities. We use Qwen-2.5-VL-3B (Bai et al., 2025) as our default backbone, considering its strong performance in both visual understanding and language generation, as well as its moderate parameter size. Based on the RoboDraw dataset, we construct three modality-specific datasets ($D_{i2s}$, $D_{t2s}$, and $D_{it2s}$) to enhance the SVG generation ability of the MLLM. Since the backbone already exhibits preliminary capabilities for generating SVGs, we apply supervised fine-tuning and reinforcement learning to further strengthen this ability. Concretely,

$$D_{i2s} = \{S_c \mid I_c\}, \quad D_{t2s} = \{S_c \mid T_c\}, \quad D_{it2s} = \{S_c \mid (I_c, T_c, G_{tc})\}, \tag{5}$$

where the notation {output | input} specifies the mapping between inputs and outputs. The corresponding three modules are Image2SVG $M_{i2v}$, Text2SVG $M_{t2v}$, and ImageText2SVG $M_{it2v}$.

Furthermore, to adapt to the PartialSVG-to-SVG task, we incorporate the partial SVG $S_p$ into the textual data and train specialized SVG generator modules, denoted as $M'_{t2s}$ and $M'_{it2s}$. The corresponding training datasets are defined as

$$D'_{t2s} = \{S_c \mid (T_c, S_p)\}, \quad D'_{it2s} = \{S_c \mid (I_c, T_c, S_p)\}, \tag{6}$$

where $M'_{t2v}$ is trained for text–PartialSVG inputs, and $M'_{it2v}$ for image–text–PartialSVG inputs.

**Inference Process.** For the four tasks defined in the RoboDraw dataset (i.e., Text-to-SVG, Image-to-SVG, PartialSVG-to-SVG, and PartialImage-to-SVG), we design task-specific workflows using the three modules of the SVG generator to produce SVG candidates. And then select the output SVG $s^k$, where $k \in \{t1, t2, t3, t4\}$.

**(i)** Text-to-SVG. Given an input text $T_c$, we first employ the text-to-image model $g_{v1}$ to generate an image guidance $G_{ic}$. The SVG generator then produces candidate SVGs using its three modules:

$$s_{i2s}^{t1} = M_{i2v}(G_{ic}), \quad s_{t2s}^{t1} = M_{t2v}(T_c), \quad s_{it2s}^{t1} = M_{it2v}(T_c, G_{ic}), \tag{7}$$

resulting in a candidate set $S_{ca}^{t1} = \{s_{i2s}^{t1}, s_{t2s}^{t1}, s_{it2s}^{t1}\}$. Although the Image2SVG module $M_{i2v}$ and the ImageText2SVG module $M_{it2v}$ are trained on images from the RoboDraw dataset, they generalize effectively during inference to the synthetic images produced by $g_{v1}$. Finally, the output SVG $s^{t1}$ is selected from $S_{ca}^{t1}$ based on the CLIPScore.

**(ii)** Image-to-SVG. The goal of the Image-to-SVG task is to generate an SVG that closely resembles the input image $I_c$, making $I_c$ an essential input for SVG generator modules. Since the Text2SVG module $M_{t2v}$ only accepts textual inputs, it is excluded from this task. Instead, we utilize the Image2SVG module $M_{i2v}$ and the ImageText2SVG module $M_{it2v}$ as

$$s_{i2s}^{t2} = M_{i2v}(I_c), \quad s_{it2s}^{t2} = M_{it2v}(I_c, G_{tc}), \tag{8}$$

where the complete description guidance $G_{tc}$ is obtained from the image captioning model $g_{t1}$ applied to $I_c$. The candidate set is then given by $S_{ca}^{t2} = \{s_{i2s}^{t2}, s_{it2s}^{t2}\}$, from which the final output $s^{t2}$ is selected based on the SSIM score.

**(iii)** PartialSVG-to-SVG. A key characteristic of the PartialSVG-to-SVG task is that the generated SVG must strictly preserve the existing partial SVG $S_p$. Therefore, we employ the specialized Text2SVG module $M'_{t2v}$ and ImageText2SVG module $M'_{it2v}$, both designed to handle inputs containing $S_p$ as

$$s_{t2s}^{t3} = M'_{t2v}(T_c, S_p), \quad s_{it2s}^{t3} = M'_{it2v}(G_{ie}, T_c, S_p), \tag{9}$$

where the edited image guidance $G_{ie}$ is obtained by first rendering $S_p$ into an image $I_p$, and then applying the image editing model $g_{v2}$ conditioned on $T_c$. As a result, $G_{ie}$ captures both the existing content in $S_p$ and the intended modifications for interactive drawing, serving as the visual guidance for the PartialSVG-to-SVG task. For the candidate set $S_{ca}^{t3} = \{s_{t2s}^{t3}, s_{it2s}^{t3}\}$, the final output SVG $s^{t3}$ is selected based on the CLIPScore.

**(iv)** PartialImage-to-SVG. The inputs for this task are a partial image $I_p$ and a complete text description $T_c$. We first apply the image editing model $g_{v2}$ to obtain an edited image guidance $G_{ie}$, and apply the suggestion making $g_{t2}$ to obtain textual suggestion $G_{tp}$. The SVG generator then employs the Image2SVG module $M_{i2v}$ and the ImageText2SVG module $M_{it2v}$ as

$$s_{i2s}^{t4} = M_{i2v}(G_{ie}), \quad s_{it2s}^{t4} = M_{it2v}(G_{ie}, T_c), \quad s'^{t4}_{it2s} = M_{it2v}(I_p, T_c, G_{tp}) \tag{10}$$

resulting in the candidate set $S_{ca}^{t4} = \{s_{i2s}^{t4}, s_{it2s}^{t4}, s'^{t4}_{it2s}\}$. The final output SVG $s^{t4}$ is then selected according to the SSIM score.

## 4 EXPERIMENTS

### 4.1 EXPERIMENTAL SETTINGS

**Dataset and Evaluation Metrics.** We utilize our constructed RoboDraw dataset for training and testing, and also adopt the SVGenius benchmark (Chen et al., 2025b) to further validate the performance of our model. In detail, RoboSVG is trained on one million samples from the Robo-Draw training split, and evaluated on 500 samples from the RoboDraw test split as well as 300 samples from the SVGenius benchmark. The evaluation metrics are divided into two categories: text-aligned SVG evaluation and image-aligned SVG evaluation. For text-aligned evaluation, we report scores of FID (Heusel et al., 2017), CLIPScore (Radford et al., 2021), Aesthetic (Schuhmann, 2022), HPS (Wu et al., 2023b), and rCLIP (Chen et al., 2025b). For image-aligned evaluation, we use scores of DINO (Oquab et al., 2024), SSIM (Wang et al., 2004), LPIPS (Zhang et al., 2018), and MSE (Goodfellow et al., 2016). Accordingly, Text-to-SVG and PartialSVG-to-SVG tasks are evaluated with text-aligned metrics, while Image-to-SVG and PartialImage-to-SVG tasks are evaluated with image-aligned metrics.

**Baselines.** We evaluate both multimodal large language models (MLLMs) and task-specific models on the RoboDraw test set. The MLLMs include Qwen2.5-VL (Bai et al., 2025) (3B and 72B

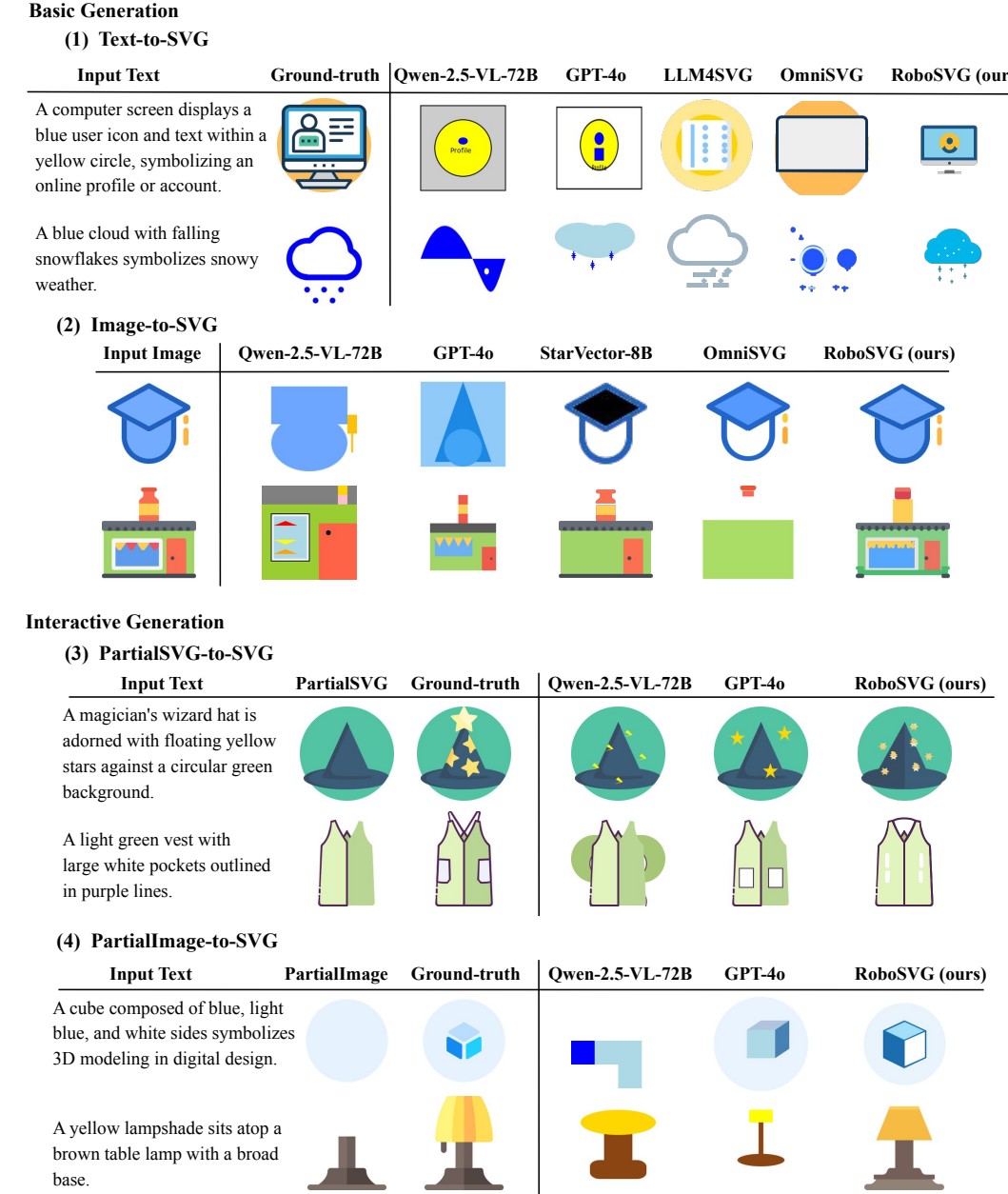

Figure 3: Qualitative results of different models on the RoboDraw test split. We illustrate four SVG generation tasks, with two examples shown for each task.

versions) and GPT-4o (Hurst et al., 2024), which are tested in a zero-shot setting for both basic and interactive SVG generation. The task-specific models include LLM4SVG (Xing et al., 2025b) (supporting Text-to-SVG), StarVector-8B (Rodriguez et al., 2025) (supporting Image-to-SVG), and OmniSVG-3B (Yang et al., 2025b) (supporting both Text-to-SVG and Image-to-SVG). In addition, the SVGenius benchmark includes results from Iconshop (Wu et al., 2023a) and Claude-3.7-Sonnet (Anthropic, 2023) for comparison.

**Implementation Details.** We primarily train the SVG generator modules based on Qwen2.5-VL-3B, using LLaMA-Factory (Zheng et al., 2024) and VLM-R1 (Shen et al., 2025) as the training framework for supervised fine-tuning and reinforcement learning. The training is conducted on the constructed RoboDraw dataset, covering both basic and interactive generation tasks. Training is carried out on 8 NVIDIA H20 GPUs with a batch size of 64, using a learning rate $1.2 \times 10^{-4}$. We

Table 1: Comparison of basic generation performance on RoboDraw test split.

| Methods | Text-to-SVG | | | | Image-to-SVG | | | |
|---|---|---|---|---|---|---|---|---|
| | FID↓ | CLIPScore↑ | Aesthetic↑ | HPS↑ | DINO↑ | SSIM↑ | LPIPS↓ | MSE↓ |
| Qwen2.5-VL-3B (Bai et al., 2025) | 165.11 | 22.11 | 3.56 | 24.15 | 0.757 | 0.480 | 0.511 | 0.213 |
| Qwen2.5-VL-72B (Bai et al., 2025) | 148.25 | 24.72 | 4.03 | 25.09 | 0.831 | 0.472 | 0.422 | 0.171 |
| LLM4SVG (Xing et al., 2025b) | 64.46 | 23.14 | 4.71 | 24.42 | - | - | - | - |
| OmniSVG-3B (Yang et al., 2025b) | 108.64 | 22.19 | 4.21 | 23.76 | 0.879 | 0.699 | 0.234 | 0.078 |
| StarVector-8B (Rodriguez et al., 2025) | - | - | - | - | 0.869 | 0.728 | 0.247 | 0.062 |
| GPT-4o (Hurst et al., 2024) | 145.67 | 26.91 | 4.26 | **25.68** | 0.862 | 0.551 | 0.364 | 0.126 |
| RoboSVG (ours) | **63.65** | **27.89** | **5.04** | 25.36 | **0.968** | **0.843** | **0.102** | **0.021** |

Table 2: Comparison of basic generation performance on SVGenius benchmark (Chen et al., 2025b).

| Methods | Text-to-SVG | | | | | | Image-to-SVG | | | | | | | | | | | |
|---|---|---|---|---|---|---|---|---|---|---|---|---|---|---|---|---|---|---|
| | easy | | medium | | hard | | easy | | | | medium | | | | hard | | | |
| | rCLIP↑ | HPS↑ | rCLIP↑ | HPS↑ | rCLIP↑ | HPS↑ | DINO↑ | SSIM↑ | LPIPS↓ | MSE↓ | DINO↑ | SSIM↑ | LPIPS↓ | MSE↓ | DINO↑ | SSIM↑ | LPIPS↓ | MSE↓ |
| Iconshop (Wu et al., 2023a) | 86.22 | 17.99 | 77.35 | 14.68 | 72.55 | 12.95 | - | - | - | - | - | - | - | - | - | - | - | - |
| LLM4SVG (Xing et al., 2025b) | 75.78 | 16.76 | 68.84 | 14.88 | 62.98 | 14.62 | - | - | - | - | - | - | - | - | - | - | - | - |
| Qwen2.5-72B (Yang et al., 2025a) | 91.18 | 17.38 | 79.91 | 16.35 | 75.22 | 15.51 | - | - | - | - | - | - | - | - | - | - | - | - |
| StarVector-8B (Rodriguez et al., 2025) | - | - | - | - | - | - | 0.7398 | 0.3760 | 0.3680 | 0.4371 | 0.6728 | 0.5295 | 0.4120 | 0.2163 | 0.6581 | 0.5653 | 0.4212 | 0.1647 |
| Qwen2.5-VL-72B (Bai et al., 2025) | - | - | - | - | - | - | 0.7983 | 0.4675 | 0.4085 | 0.2452 | 0.7641 | 0.4529 | 0.4793 | 0.2004 | 0.7588 | 0.4274 | 0.4986 | 0.1772 |
| GPT-4o (Hurst et al., 2024) | 90.95 | 20.35 | 84.72 | 17.51 | 82.81 | 16.69 | 0.8748 | 0.5241 | 0.3381 | 0.1921 | 0.8129 | 0.4941 | 0.4229 | 0.1544 | 0.8136 | 0.5066 | 0.4340 | 0.1277 |
| Claude-3.7-Sonnet (Anthropic, 2023) | 92.90 | 21.35 | 87.62 | 18.57 | 87.97 | 18.74 | 0.8991 | 0.5402 | 0.3212 | 0.1713 | 0.8617 | 0.5136 | 0.3767 | 0.1233 | 0.8551 | 0.5115 | 0.3959 | 0.1123 |
| RoboSVG (ours) | 95.96 | 25.45 | 95.08 | 25.88 | 94.31 | 25.54 | 0.9853 | 0.8852 | 0.0542 | 0.0193 | 0.9456 | 0.7241 | 0.1709 | 0.0433 | 0.9375 | 0.6692 | 0.2051 | 0.0493 |

Table 3: Comparison of interactive generation performance on RoboDraw test split.

| Methods | PartialSVG-to-SVG | | | | PartialImage-to-SVG | | | |
|---|---|---|---|---|---|---|---|---|
| | FID↓ | CLIPScore↑ | Aesthetic↑ | HPS↑ | DINO↑ | SSIM↑ | LPIPS↓ | MSE↓ |
| Qwen2.5-VL-3B (Bai et al., 2025) | 162.04 | 23.34 | 4.00 | 24.47 | 0.752 | 0.456 | 0.512 | 0.232 |
| Qwen2.5-VL-72B (Bai et al., 2025) | 145.59 | 25.30 | 4.29 | 25.17 | 0.818 | 0.501 | 0.436 | 0.143 |
| GPT-4o (Hurst et al., 2024) | 144.57 | 26.66 | 4.57 | **25.55** | 0.846 | 0.541 | 0.394 | 0.129 |
| RoboSVG (ours) | **50.20** | **27.10** | **5.00** | 25.20 | **0.892** | **0.659** | **0.288** | **0.075** |

Table 4: SVG generation modules performance on RoboDraw basic generation test split.

| Methods | Text-to-SVG | | | | Image-to-SVG | | | |
|---|---|---|---|---|---|---|---|---|
| | FID↓ | CLIPScore↑ | Aesthetic↑ | HPS↑ | DINO↑ | SSIM↑ | LPIPS↓ | MSE↓ |
| Image2SVG module | 86.45 | 27.01 | 4.96 | 25.28 | 0.966 | 0.828 | 0.108 | 0.024 |
| Text2SVG module | **60.43** | 25.26 | 4.97 | 25.05 | - | - | - | - |
| ImageText2SVG module | 87.02 | 26.64 | 4.90 | 25.20 | 0.960 | 0.802 | 0.131 | 0.031 |
| RoboSVG (full model) | 63.65 | **27.89** | **5.04** | **25.36** | **0.968** | **0.843** | **0.102** | **0.021** |

Table 5: SVG generation modules performance on RoboDraw interactive generation test split.

| Methods | PartialSVG-to-SVG | | | | PartialImage-to-SVG | | | |
|---|---|---|---|---|---|---|---|---|
| | FID↓ | CLIPScore↑ | Aesthetic↑ | HPS↑ | DINO↑ | SSIM↑ | LPIPS↓ | MSE↓ |
| Image2SVG module | - | - | - | - | 0.886 | 0.617 | 0.324 | 0.088 |
| Text2SVG module | **50.92** | 25.58 | 4.99 | 25.12 | - | - | - | - |
| ImageText2SVG module | 61.40 | 25.59 | 4.83 | 25.01 | 0.877 | 0.534 | 0.369 | 0.133 |
| RoboSVG (full model) | 63.65 | **27.89** | **5.04** | **25.36** | **0.892** | **0.659** | **0.288** | **0.075** |

employ the AdamW optimizer (Loshchilov & Hutter, 2019) with a cosine learning rate scheduler, using 1,000 warm-up iterations and train for 5 epochs.

## 4.2 RESULTS FOR SVG GENERATION

Here, we analyze the performance of RoboSVG on SVG generation tasks. We begin by presenting the overall evaluation results on RoboDraw and SVGenius datasets, followed by ablation studies to examine the roles of multimodal guidance and the SVG generation modules. We then analyze the qualitative results and conduct a user study to further compare the capabilities of different models.

**Main Results.** The overall performance of the RoboSVG model on SVG generation tasks is summarized in Tables 1 to 3. Thanks to the incorporation of multimodal guidance and well-trained SVG generation modules, RoboSVG achieves significant improvements over comparison approaches. Table 1 reports results on basic generation (Text-to-SVG and Image-to-SVG) on the RoboDraw test split. In the Text-to-SVG task, RoboSVG attains a CLIPScore of 27.89, surpassing the baseline Qwen2.5-VL-3B by +5.78, demonstrating the effectiveness of our model design. In the Image-to-SVG task, RoboSVG outperforms the second-best model, StarVector-8B, across all metrics (e.g., SSIM 0.843 vs. 0.728). We attribute this substantial gain to the simplified and standardized design of the RoboDraw dataset, which provides cleaner supervision for training. Table 2 compares different models on the SVGenius benchmark, where RoboSVG again achieves the best results by a clear margin (e.g., rCLIP 93.70, MSE 0.019 for SVGenius easy split), demonstrating its strong generalization. Results on interactive SVG generation are reported in Table 3. RoboSVG achieves the best performance on 7 out of 8 metrics compared to strong MLLM baselines such as Qwen2.5-VL-72B and GPT-4o. The only exception is HPS, where GPT-4o outperforms RoboSVG slightly. We attribute this to stylistic differences: GPT-4o tends to generate simpler SVGs, while RoboSVG often produces more cartoon-like outputs mimicking the distribution of the RoboDraw training split.

Table 6: Image2SVG module performance on Image-to-SVG task under different training conditions. Here, 3B and 7B denote the parameter size of backbone Qwen-2.5-VL; 200 and 224 denote the input image resolution.

| Conditions | | Image-to-SVG | | | |
|---|---|---|---|---|---|
| Parameter | Resolution | DINO↑ | SSIM↑ | LPIPS↓ | MSE↓ |
| 3B | 200 | 0.937 | 0.734 | 0.172 | 0.062 |
| 3B | 224 | 0.966 | 0.828 | 0.108 | 0.024 |
| 7B | 224 | **0.972** | **0.828** | **0.102** | **0.023** |

Table 7: User study on different methods. Here, QC, VQ, and PU denote Query Compliance, Visual Quality, and Practical Usage, respectively.

| Methods | Text-to-SVG | | | Image-to-SVG | | | PartialSVG-to-SVG | | | PartialImage-to-SVG | | |
|---|---|---|---|---|---|---|---|---|---|---|---|---|
| | QC↑ | VQ↑ | PU↑ | QC↑ | VQ↑ | PU↑ | QC↑ | VQ↑ | PU↑ | QC↑ | VQ↑ | PU↑ |
| Qwen-2.5-VL-72B (Bai et al., 2025) | 0.17 | 0.18 | 0.04 | 0.15 | 0.23 | 0.07 | 0.10 | 0.12 | 0.01 | 0.06 | 0.15 | 0.02 |
| GPT-4o (Hurst et al., 2024) | 0.52 | 0.42 | 0.18 | 0.14 | 0.27 | 0.04 | 0.49 | 0.45 | 0.21 | 0.36 | 0.29 | 0.06 |
| RoboSVG (ours) | **0.69** | **0.69** | **0.48** | **0.80** | **0.62** | **0.55** | **0.50** | **0.52** | **0.28** | **0.54** | **0.65** | **0.47** |

**Ablation studies.** We first analyze the impact of different SVG generation modules. Table 4 reports results on basic generation tasks, while Table 5 presents results on interactive generation tasks. As shown, the RoboSVG (full model) consistently outperforms any single module. This improvement arises from the numerical guidance, which selects the best output SVG from multiple module-generated SVG candidates during inference. Among the three modules, the Image2SVG module achieves the strongest performance (e.g., CLIPScore 27.01 on Text-to-SVG, DINO 0.966 on Image-to-SVG). We attribute the outperformance of the Image2SVG module to two factors: **(i)** the trained MLLM effectively captures the image-to-SVG mapping defined in RoboDraw; **(ii)** the supporting text-to-image and image editing models provide high-quality visual guidance that accurately represent the intended target. Training conditions have a clear impact on SVG generation performance. We analyze this effect using the Image2SVG module on the Image-to-SVG task, with results summarized in Table 6. Increasing the image resolution from 200×200 to 224×224 yields a notable improvement (DINO from 0.966 to 0.937), as the higher resolution provides richer visual information and aligns better with the native input size of Qwen-2.5-VL. Scaling the backbone from 3B to 7B parameters yields only a slight gain (DINO increases from 0.966 to 0.972) while incurring a substantially higher computational cost. Therefore, our default experiments are conducted with Qwen-2.5-VL-3B for efficiency.

**Qualitative Results.** For the four proposed tasks, we select two representative cases per task to compare the performance of different methods, as shown in Figure 3. Overall, RoboSVG better satisfies the input queries and achieves superior visual quality compared to other methods. For example, in the Text-to-SVG task, RoboSVG generates "computer screen" and "snowflakes" that are faithful to real-world appearances. In the Image-to-SVG task, the generated mortarboard accurately matches the shape and color of the input image. RoboSVG is able to preserve the key visual characteristics even for complex cases, such as a house drawing. More qualitative results of RoboSVG can be seen in Appendix C.

**User Study.** We conduct a user study in which participants evaluate each method on three dimensions (i.e., query compliance, visual quality, and practical usage) using 0/1 binary scores (see Appendix B for more details). The aggregated results are reported in Table 7. RoboSVG achieves the highest scores across all evaluation dimensions. Notably, in the Image-to-SVG task, RoboSVG achieves a query compliance score of 0.80, representing a substantial improvement over GPT-4o (0.14), which highlights RoboSVG's strong capability in handling image-conditioned SVG generation. In the PartialSVG-to-SVG task, RoboSVG only slightly outperforms GPT-4o. This highlights the complexity of the PartialSVG-to-SVG task, which requires both a strong semantic understanding and precise generation capabilities.

## 5 CONCLUSION

In this work, we explore SVG generation as a paradigm for interactive robot drawing. We first curated the RoboDraw dataset, a large-scale dataset that defines four SVG generation tasks: Text-to-SVG, Image-to-SVG, PartialSVG-to-SVG, and PartialImage-to-SVG. Building on this dataset, we developed the RoboSVG model, a unified framework that leverages multimodal guidance to enhance SVG generation. Experimental results demonstrate that the RoboSVG model excels across all four tasks, achieving substantial improvements over strong baselines (e.g., GPT-4o and OmniSVG).

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

# Supplementary Material for RoboSVG: A Unified Framework for Interactive SVG Generation with Multi-modal Guidance

We first present the prompts used in RoboSVG, then provide further details of the user study, and finally analyze additional qualitative results.

## A    PROMPTS

Table 8: Prompts on different settings (including guidance generator and SVG generator).

| Settings | Prompts |
|---|---|
| Text-to-Image $g_{v1}$ | Minimalist vector-style icon of {description}. Empty background. |
| Image Editing $g_{v2}$ | Keep as many original elements as possible, but edit by adding elements to transform it into a minimalist vector-style icon: {description} |
| Image Captioning $g_{t1}$ | Please describe it within 50 words. |
| Image2SVG module $M_{i2v}$ | Convert this raster image to SVG code. |
| Text2SVG module $M_{t2v}$ | Generate an SVG illustration from the given description: {description} |
| ImageText2SVG module $M_{it2v}$ | Convert this raster image to SVG code with the following description: {description} |
| ImageText2SVG module (PartialSVG) $M'_{it2v}$ | Please complete the SVG code so that it fully represents the following description. Make sure to include the existing SVG code in the final result. Description: {description}. Existing SVG code: {partial SVG} |

Table 8 presents the prompts used in RoboSVG under different settings. The first three are applied within the Guidance Generator, while the following four are used in the SVG Generator. In the table, {description} denotes the complete description $T_c$, and {partial SVG} denotes the partial SVG code $S_p$.

## B    DETAILS OF USER STUDY

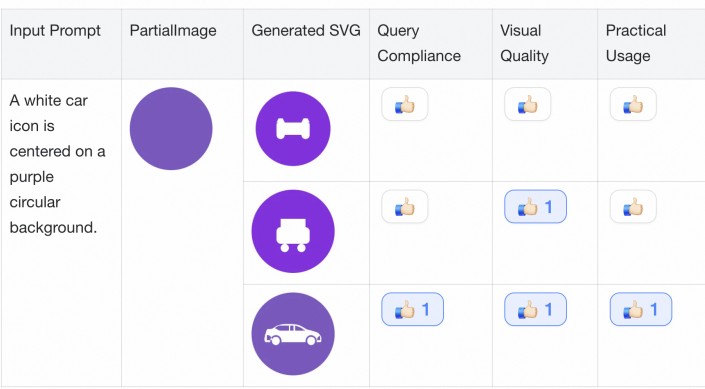

Figure 4: The interface example of the user study. Here, we present an example of the PartialImage-to-SVG task, where the input query consists of the input prompt and PartialImage, and the generated SVGs are produced by three different methods (Qwen-2.5-VL-72B, GPT-4o, and our RoboSVG).

We conduct a user study on the four proposed tasks, evaluating along three dimensions (query compliance, visual quality, and practical usage) in Figure 4. Similar to those in LLM4SVG (Xing et al., 2025b), query compliance measures whether the generated SVG matches the input query, visual quality assesses the plausibility of the rendered output, and practical usage evaluates its applicability

in real scenarios. The compared methods include Qwen-2.5-VL-72B, GPT-4o, and our RoboSVG. For each task, we randomly sample 10 cases, and 10 users independently evaluate the outputs of all methods, resulting in a total of 3,600 judgments.

## C    MORE QUALITATIVE RESULTS

**Basic Generation**

**(1) Text-to-SVG**

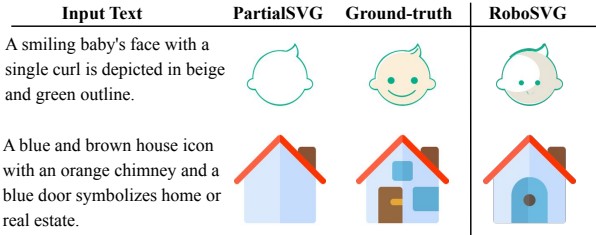

**(2) Image-to-SVG**

**Interactive Generation**

**(3) PartialSVG-to-SVG**

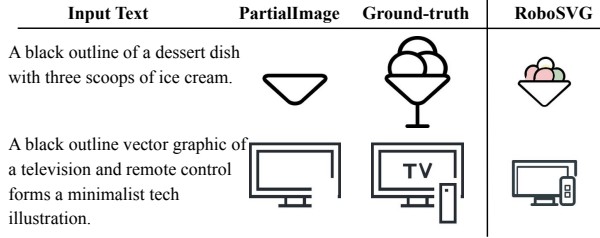

**(4) PartialImage-to-SVG**

Figure 5: Qualitative results of RoboSVG on four SVG generation tasks.

Figure 5 presents additional qualitative results of RoboSVG. Overall, the generated SVGs align well with the input queries, e.g., the "green tree" (fourth case in Text-to-SVG) and the "blue and brown house" (second case in PartialSVG-to-SVG). Nevertheless, there remains space for improvement in finer details, such as rendering the "%" symbol in the pie chart (third case in Image-to-SVG).

