# OpenReview forum: "RoboSVG: A Unified Framework for Interactive SVG Generation with Multi-modal Guidance"
_ICLR.cc/2026/Conference — ICLR 2026 Conference Withdrawn Submission_

### Official Review · Reviewer_7gWu · 2025-10-28

**Soundness:** 2
**Presentation:** 2
**Contribution:** 2
**Rating:** 4
**Confidence:** 3

**Summary:**

This paper introduces RoboSVG, a novel family of SVG generation models capable of producing vector graphics not only from text prompts but also from full or partial raster images. RoboSVG enables interactive SVG generation, supporting tasks such as PartialSVG-to-SVG and PartialImage-to-SVG. To facilitate research in this direction, the authors introduce MMSVG-2M, a large-scale multimodal dataset tailored for these tasks, along with RoboDraw, a new benchmark for evaluating SVG generation quality.

In practice, RoboSVG enhances multimodal inputs through augmentation techniques and concurrently employs three specialized models—text-to-SVG, image-to-SVG, and text-image-to-SVG. The final output is selected from their results based on predefined quality criteria.

Comprehensive evaluations demonstrate that RoboSVG achieves state-of-the-art (SOTA) performance across all tasks on both the SVGenius and RoboDraw benchmarks, outperforming existing methods.

**Strengths:**

1. This work provides a clear and precise definition of the proposed interactive generation framework.
2. It explicitly defines the different task variants (e.g., PartialSVG-to-SVG and PartialImage-to-SVG), enhancing clarity and reproducibility.
3. The method is evaluated from multiple perspectives and rigorously compared against several prior approaches, demonstrating its effectiveness and versatility.

**Weaknesses:**

My main concerns relate to the experimental design and the fairness of comparisons
1. **Ensemble fairness**: Your method uses three SVG generation modules and selects the best output via numerical guidance. Have you evaluated whether baseline models (e.g., GPT-4o, Qwen2.5-VL) also benefit from multi-trial generation (e.g., running 2–3 times with different seeds and selecting the best)? Without this, the comparison may be biased in favor of your pipeline.
2. **Two-stage generation for partial tasks**: Since SVGs are inherently sequential and code-like, a two-stage approach—first generating a partial code of SVG from the partial input (e.g., image), then completing the code—might be more suitable for general MLLMs. Have you compared such a strategy?
3. **Missing OmniSVG on SVGenius**: OmniSVG is a strong SVG-specific baseline and appears to be SOTA in prior work, yet its results are absent from the SVGenius benchmark. Including it would strengthen the validity of your claims.
4. **Use of FID on RoboDraw**: FID measures distribution-level similarity and is sensitive to dataset bias. It may unfairly penalize models that generate high-quality but stylistically different outputs.

**Questions:**

1. The authors mention generating textual suggestions to guide SVG generation—could you provide an example for reference?
2. In the data construction part, you mention the tuple of (Tc, Sc, Sp), but I think it is important to show how you select the Sp. In addition, I wonder whether it is possible to generate several Sp from one sample? And would this enhance the generalization of the model?

---

### Official Review · Reviewer_89yc · 2025-10-30

**Soundness:** 2
**Presentation:** 2
**Contribution:** 2
**Rating:** 4
**Confidence:** 4

**Summary:**

This paper introduces RoboSVG, a unified framework for generating Scalable Vector Graphics (SVGs) from multi-modal inputs. The authors contribute: (1) RoboDraw, a dataset of 1 million SVG samples with text descriptions and partial SVGs; (2) four SVG generation tasks including basic generation (Text-to-SVG, Image-to-SVG) and interactive generation (PartialSVG-to-SVG, PartialImage-to-SVG); and (3) a two-stage framework consisting of a guidance generator that produces multi-modal guidance and an SVG generator with three specialized modules. Experiments show improvements over baselines including GPT-4o and task-specific models.

**Strengths:**

1. Introduces interactive SVG generation tasks (PartialSVG-to-SVG, PartialImage-to-SVG) which are novel and practical
2. RoboDraw dataset enables systematic study of these tasks
3. First comprehensive benchmark comparing MLLMs and specialized models on SVG generation

**Weaknesses:**

1. Limited technical novelty - the approach primarily combines existing components (Qwen-2.5-VL backbone, FLUX.1 for guidance, etc.)
2. RoboDraw is constructed from existing datasets (MMSVG-2M, SVGX) with filtering and processing, not entirely original data collection
3. The "unified framework" is essentially task-specific modules with candidate selection, which is relatively straightforward
4. No significant algorithmic innovations beyond engineering existing techniques

**Questions:**

1. The approach is primarily an engineering effort combining existing models (Qwen-2.5-VL, FLUX.1, etc.) without significant algorithmic innovations. The "multimodal guidance" is essentially using off-the-shelf tools in a pipeline.
2. Built on existing datasets (MMSVG-2M, SVGX), so contribution is primarily curation/filtering
3. Metrics like CLIPScore may not fully capture SVG-specific qualities (structural validity, path efficiency, editability)
4. No evaluation of SVG code quality (e.g., number of paths, code complexity)
5. What is the end-to-end inference time and computational cost? How does this compare to baselines? The pipeline involves text-to-image generation, multiple MLLM forward passes, etc.
6. Have you considered SVG-specific quality metrics like path efficiency, code compactness, or editability?
I am willing to engage in detailed discussion with the authors during the rebuttal phase.

---

### Official Review · Reviewer_MqLe · 2025-10-31

**Soundness:** 3
**Presentation:** 4
**Contribution:** 2
**Rating:** 2
**Confidence:** 5

**Summary:**

This paper propose RoboSVG, a system for interactive SVG generation from multimodal inputs like text, images, and partial SVGs.
This framework first uses a guidance generator to process the input. The results are fed into a fine-tuned MLLM to produce multiple candidates. Finally, a simple metric is used to select the best output.

**Strengths:**

1. The paper is very well presented keeping minute details in mind.
2. The paper clearly defines a practical and useful problem of interactive SVG generation which is a logical step beyond one-shot static generation.
3. Experiments shows system achieves strong empirical results, consistently outperforming strong zero-shot baselines and existing specialized models on their new benchmark.

**Weaknesses:**

1. The paper presents a system, not a novel method. This seems more like an engineering system rather than a well-defined mathematical formulation.
2. The main baselines, GPT-4o and Qwen-72B, are run in a zero-shot setting. RoboSVG is fine-tuned on 1M samples from the RoboDraw dataset. This is an apples-to-oranges comparison. The specialized, fine-tuned model will win in any scenario here.

**Questions:**

1. To create a fair comparison, what is the performance of the baseline models when fine-tuned on the exact same RoboDraw training split?
2. What is the performance of the fine-tuned SVG generator modules on their own, without the multi-candidate generation and re-ranking?
3. How does the system handle OOD requests that don't look like the clean, icon-style graphics from the RoboDraw dataset?

---

### Official Review · Reviewer_XkHv · 2025-11-03

**Soundness:** 3
**Presentation:** 3
**Contribution:** 2
**Rating:** 4
**Confidence:** 3

**Summary:**

This paper introduces a new curated dataset and a framework for SVG generation. The main contributions are threefold:

1. **Introducing a new task**: The authors introduce interactive SVG generation as a new task, taking a partial SVG or an image representing a partial shape and generating a complete SVG. For this new task, the authors provide partial–full pairs in the dataset.

2. **Curated dataset**: While the authors source data from existing datasets (MMSVG-2M and SVGX), they further curate these datasets by selecting only high-quality samples whose text descriptions are highly aligned with the corresponding SVGs.

3. **Generative framework (RoboSVG)**: The proposed generative model, RoboSVG, consists of a guidance generator that produces guidance from one conditioning modality to another (e.g., generating an image from text or text from an image). The authors also introduce a reward function based on numerical guidance tools, which is used for reinforcement learning fine-tuning. Given the conditioning input and generated guidance, the Qwen model is used as the backbone and fine-tuned to generate SVGs.

The experimental results demonstrate that the proposed generative framework performs well and outperforms other pretrained foundational models.

**Strengths:**

- The paper is clearly written. The technical contributions and key ideas are easy to follow.

- The authors present comprehensive analyses across diverse SVG generation tasks, including different conditioning inputs and cases where partial SVGs or input images are present or absent.

- The user study provides useful evidence supporting the superior performance of the proposed method.

**Weaknesses:**

- While the paper reads somewhat like a positioning paper introducing the interactive SVG generation task, its novelty is limited: (1) SVG generation has already been studied in prior work, (2) the dataset is a curated version of existing datasets rather than newly collected, and (3) the method relies heavily on existing foundation models.

- The paper lacks clear technical novelty or methodological contributions. While such positioning papers may fit better in NLP venues, I think ICLR generally expects a higher degree of technical innovation. The proposed framework mainly involves using and fine-tuning existing models with existing techniques.

- Some important technical details are missing. For example:

  - Although the authors mention that numerical guidance tools are used as rewards for reinforcement learning (lines 235–236), the paper does not clearly explain how they are incorporated into the reward function.

  - The authors emphasize interactive generation as a new task but do not detail how the partial inputs are constructed in the dataset. For a task aiming at interactive generation, the partial input should ideally correspond to user-driven incremental updates rather than random subsets. These details are not described. Moreover, if the authors highlight interactive generation as a major contribution, the user study should also include evaluation on interactive generation scenarios.

**Questions:**

In lines 154–155, are the drawing commands {M, L, C, Q, A, Z}?

---

### Note · Authors · 2025-11-24

**Comment:**

Thank you for the reviewers’ valuable comments and suggestions. They have been very helpful in guiding our further research. Thank you.

**Withdrawal Confirmation:**

I have read and agree with the venue's withdrawal policy on behalf of myself and my co-authors.